# Recent Advances in Functional Nanomaterials for Diagnostic and Sensing Using Self-Assembled Monolayers

**DOI:** 10.3390/ijms241310819

**Published:** 2023-06-28

**Authors:** Caroline R. Basso, Bruno P. Crulhas, Gustavo R. Castro, Valber A. Pedrosa

**Affiliations:** Institute of Bioscience, UNESP, Botucatu 18618-000, SP, Brazil

**Keywords:** self-assemblies, nanostructures, sensor

## Abstract

Functional nanomaterials have attracted attention by producing different structures in any field. These materials have several potential applications, including medicine, electronics, and energy, which provide many unique properties. These nanostructures can be synthesized using various methods, including self-assembly, which can be used for the same applications. This unique nanomaterial is increasingly being used for biological detection due to its unique optical, electrical, and mechanical properties, which provide sensitive and specific sensors for detecting biomolecules such as DNA, RNA, and proteins. This review highlights recent advances in the field and discusses the fabrication and characterization of the corresponding materials, which can be further applied in optical, magnetic, electronic, and sensor fields.

## 1. Introduction

Self-assembly monolayers (SAMs) are robust processes in which individual components spontaneously organize themselves into a functional whole without external intervention [1]. This process can occur at different length scales, from the microscopic to the macroscopic. It can create various structures and materials, including photonic crystals, quantum dots, and supramolecular assemblies [2,3]. Intermolecular forces, such as hydrophobic interactions, ionic interactions, van der Waals forces, hydrogen bonding, covalent bonding, and coordination bonding, drive the self-assembly process [4]. These forces can be used to create complex networks and other structures with novel properties that operate at the nanoscale, resulting in self-organized nanostructures. Jean-Marie Lehn, Donald Cram, and Charles Pedersen were awarded the Nobel Prize in Chemistry in 1987 for developing and using molecules with structure–specific interactions of high specificity and selectivity. They elucidated the factors that determined the ability of the molecules to recognize each other and fit into one another like a key fits a lock resulting in self-assembly properties only by changing the specific part of the structure of the molecular building blocks used [5,6].

Supramolecular chemistry is a branch of chemistry that uses these intermolecular forces to design and synthesize complex molecular assemblies with well-defined structures and functions. Supramolecular chemistry offers a versatile and scalable approach to nanomanufacturing, enabling the fabrication of nanostructured materials and devices with high precision and control. By manipulating intermolecular forces, supramolecular chemists can design and synthesize structures with unique properties, such as a high surface area, tunable electronic properties, and specific chemical reactivity. One example of supramolecular chemistry in action is the creation of self-assembled monolayers (SAMs), which are the spontaneous organization of molecules on a substrate surface form. Due to their unique size-dependent properties and potential applications in various fields, including biomedical engineering and regenerative medicine, SAMs have been studied. Additionally, they can be made from various materials, including metals, semiconductors, and polymers, and assembled into various shapes and sizes. Their properties can be tuned by varying the composition of the material and the intermolecular forces that drive the self-assembly process. Overall, the self-assembly process and supramolecular chemistry offer a promising avenue for designing and fabricating functional nanomaterials and devices with precise control over their size, shape, and properties. These approaches have the potential to revolutionize various fields, including electronics, energy, and biomedicine, by enabling the development of novel materials and devices with enhanced performance and functionality.

The main objective of this review was to provide an overview of the state-of-the-art concerning the current evolution of this field from fundamental to applied investigations and its applicability in chemical and biological sensors. This review starts with an introduction to different approaches that can synthesize nanostructures and their technological implications. A detailed discussion on the physics and chemicals of self-assembly follows this. We highlight most nanostructures used in diagnostic and sensing using SAMs. Some final considerations are then made to determine the following challenges in this area of research. This review constitutes a valuable database for non-specialist readers who wish to obtain an overview of the current trends related to self-assembled functional nanomaterials. 

## 2. Self-Assembled Nanomaterials

The synthesis of nanomaterials and the organization of structures into ordered functional and operational arrangements are crucial aspects in the field of nanotechnology since the expectation is to obtain a uniform product (with mastery of morphology, size, and particle size distribution) that is reproducible and whose physical and chemical properties are controlled over a wide range of sizes and compositions. However, developing simple and versatile methods for the preparation of nanoparticles with these characteristics is a challenging task, subordinated to the rigorous reproducibility of experimental synthesis conditions.

Over the last decade, self-assembled (SAMs) have attracted significant attention due to their unique size-dependent properties and potential applications in various fields [7,8,9]. In particular, SAMs have shown promise in biomedical engineering and regenerative medicine due to their ability to control cell behavior and promote tissue regeneration. They can be made from various materials, including metals, semiconductors, and polymers [10,11], and assembled into various shapes and sizes. Their properties can be tuned by varying the composition of the material. SAMs are typically composed of two or more different types of molecules, which self-assemble into a defined structure [12,13]. This self-assembly process is driven by forces, including hydrophobic interactions, electrostatic interactions, and van der Waals forces [14,15,16]. These several unique properties make them attractive for biomedical applications. SAMs can be designed to have a variety of shapes and sizes, allowing them to be used in various applications, such as drug delivery, tissue engineering, and biosensing [17]. Second, SAMs can be made from biocompatible materials, making them safe for humans [18]. Third, SAMs could be functionalized with various molecules, such as proteins and DNA, which allows them to interact with cells and tissues in a specific way [19]. Finally, they have been assembled into 3D structures, enabling them to mimic the structure and function of native tissues. They have been used in various biomedical applications, including drug delivery, tissue engineering, and biosensing [20]. Drug delivery is a particularly promising application for SAMs and has been used to target specific cells and tissues with therapeutic agents, such as drugs, proteins, and DNA [20]. Tissue engineering is another potential application in which SAMs can create 3D scaffolds for tissue regeneration and be functionalized with molecules to promote cell growth and differentiation.

Nevertheless, this review focuses on biosensing applications. We introduce new methodologies that can detect biomolecules, including proteins, DNA, and small molecules, which have great potential for future biomedical applications. However, a few challenges must be addressed before functional nanomaterials can be used clinically. First, they need to be made from biocompatible materials. Second, they need to be functionalized with specific molecules to interact with cells and tissues in a specific way, and third, they need to be assembled into 3D structures. Finally, testing needs to be conducted in clinical trials.

### 2.1. Diagnostics and Sensing

#### Magnetic Sensors

Functional magnetic sensors are devices that can detect and measure microscopic magnetic fields. These devices are used in various applications, including medical imaging, navigation, and security. Functional magnetic sensors are typically made from materials that are sensitive to magnetic fields, such as certain metals or semiconductors [21]. When a magnetic field is applied to these materials, it causes a change in the material’s electrical properties. This change can be detected and measured, allowing the magnetic field to be mapped [22]. Functional magnetic sensors have a wide range of applications. In medicine, they are used for magnetic resonance imaging (MRI), which allows doctors to see inside the human body without surgery. Functional magnetic sensors are essential for many modern technologies. They are precise, sensitive, and non-invasive, making them ideal for various applications.

##### Iron Nanoparticle

Magnetic nanostructures can be made from various materials, including metals and semiconductors. Iron nanoparticles are attractive for use in their field due to their unique magnetic and optical properties [23,24]. For example, they can be used to create magnetic field sensors; when light is exposed, iron nanoparticles can absorb and store energy, which can then be released as heat. This makes them ideal for use in thermal sensors. Iron nanoparticles can also create novel sensing platforms that combine optical and magnetic sensing.

Since 2010, several studies with iron oxide combined with carbon nanostructures and graphene oxide (GO) have demonstrated high electrocatalytic efficiency and have been applied to different analytes [25,26,27,28,29,30]. The idea behind these methodologies is to use the carbon nanostructure as a support to ensure size uniformity, which prevents agglomeration and creates a better surface–volume ratio. It can also be noted that including these carbon nanostructures promotes high mechanical strength, an excellent surface area, and improved electron transfer, increasing thermal and electrical conductivities, and stimulating better electrocatalytic activity [31].

Fluorescent organic nanoparticles have also attracted increasing attention for chemical or biological sensing and imaging due to their low toxicity, facile fabrication, and surface functionalization [32,33]. Recently Yang et al. designed water-soluble fluorophore binding with adenosine-5′-triphosphate (ATP), which showed strongly enhanced fluorescence. These fluorescent nanoparticles exhibited high sensitivity and selectivity toward Fe^3+^ sensing with a detection limit of 0.1 nM. In addition, after incubation with HeLa cells, the fluorophore showed excellent imaging performance by interaction with entogenous ATP in cells. This new methodology was demonstrated capable of Fe^3+^ sensing via fluorescence quenching in the cellular environment [34]. Wang et al. synthesized a small-molecule compound with triazole as a core and benzothiadiazole derivative as the branches. Combining the σ donor (nitrogen lone-pair electrons) and π receptor (highly electron-deficient aromatic ring) made triazoles a suitable ligand for Fe^3+^. This methodology showed high selectivity and sensitivity to Fe(III) ions based on fluorescence quenching in the presence of other competing metal ions [35].

##### Self-Assembling Paramagnetic

Paramagnetic particles have been used as a model system to investigate the field-induced assembly in dipolar systems [36,37]. In addition to typical colloidal interactions, paramagnetic particles acquire a dipole moment in an external magnetic field. Due to the alignment of superparamagnetic nano-domains, typically with a radius <50 nm, they are surrounded by a non-magnetic material that makes up the particle’s bulk [38,39,40,41]. When the paramagnetic material is close to another paramagnetic material, it aligns itself with the other material. This alignment allows the sensor to detect the presence of the paramagnetic material caused by the spin of electrons in the paramagnetic material. Recently, a highly selective and sensitive chemosensor for Fe^3+^ ions using 2-(4-methylbenzo-[d]thiazole-2-ylimino)methyl)phenol was developed [42]. The strong binding of Fe^3+^ ions with the hydroxyl group, imino group, and thiazole nitrogen was proposed, resulting in enhanced fluorescence. A charge transfer mechanism operates, resulting in blue shift and fluorescence enhancement. Hong et al. proposed a magnetic relaxation sensing method based on gold nanoparticles (Au NPs)-assisted triple self-assembly cascade signal amplification for the sensitive detection of aflatoxin (AFB1) [43]. AFB_1_ antibody and initiator DNA (iDNA) were labeled on Au NPs to form an Ab-Au-iDNA probe. The Fe^3+^ solution, providing paramagnetic ions with a strong magnetic signal, could be adsorbed by the polydopamine due to the formation of coordination bonds of phenolic hydroxyl groups with Fe^3+^. This practical interaction between polydopamine and Fe^3+^ significantly changed the transverse relaxation time signal of the Fe^3+^ supernatant solution, which could be used as a magnetic probe for the sensitive detection of AFB_1_. The sensor exhibited high specificity and sensitivity with a detection limit of 0.453 pg/mL. Another methodology showed a new highly selective magnetic relaxation-based method for the screening potential of Sortase A inhibitors [44]. A 13-amino acid-long peptide substrate of sortase A was conjugated to SiO_2_-EDTA-Gd NPs. In the presence of sortase A, the LPXTG motif on the peptide strand was cleaved, resulting in a shortened peptide and a reduced water T2 value whose magnitude depended on the concentration of sortase A. The detection limit was found to be 76 pM. This method was successfully applied to detect sortase A activity in bacterial suspensions. The feasibility of screening different inhibitors was tested in *Escherichia coli* and *S. aureus suspensions*.

## 3. Electrochemical and Amperometric Sensors

### 3.1. Gold Nanoparticles

Gold nanoparticles (AuNPs) are one of the most used functional nanomaterials for biosensing applications due to their unique optical, electronic, thermal, and catalytic characteristics, which can be applied in several fields, such as physics, chemistry, biology, medicine, biotechnology, and material science [45,46,47,48]. The characterization of AuNPs is a well-exploited field due to the discoveries made by Brust et al. and Schimd et al. that paved the way for developing several synthesis methods for sensing applications [46,49].

Usually, AuNP fabrication relies on the chemical reduction in gold salts in aqueous, organic phases or two phases. However, due to their high surface energy, AuNPs are overly reactive, which can lead to the aggregation of nanoparticles without protection on their surface. In this way, a stabilizer agent binds to their surface and stabilizes AuNP synthesis. It also enables the synthesis of a range of shapes of AuNPs (Figure 1). The most common methods to passivate the surface are thiol-functionalized organics, microemulsions, dispersion in polymeric matrices, and self-assembly monolayers [46,50,51,52].

The AuNPs synthesis can be divided into three subcategories:(a)*Chemical methods:* AuNPs are synthesized in an aqueous solution with a reducing agent such as citrate and sodium borohydride. The most common chemical method was developed by Turkevich, which provided stable colloidal AuNPs 15 to 50 nm in diameter. Another critical method for chemical synthesis is the one developed by Hauser et al., which standardized the Turkevich synthesis. Additionally, Brust et al. used thiol ligands to protect AuNPs [53,54].(b)*Biological methods:* AuNPs can also be synthesized by green chemistry, leading to a reduction in toxic waste that was generated during the procedure; in this way, several compounds isolated from plants, bacteria, algae, and viruses can be used to produce AuNPs with different sizes and shapes [55,56].(c)*Physical methods:* The most common physical methods are γ-radiation, ultraviolet, laser ablation, microwave, and irradiation. For example, γ-radiation provides ultrapure AuNPs in the 5 to 40 nm diameter range. In parallel, the ultraviolet method combined with high temperature can provide AuNPs of several sizes [57,58].

Moreover, the exclusive properties of nanostructured AuNPs support the manufacturing of novel biological recognition-interfaced materials with enhanced biocompatibility and a particular structure, magnetic, optic, and electronic features for biosensing applications; in this review, it is demonstrated how different sensing approaches can be performed by combining AuNPs and self-assembled monolayers [46,47].

### 3.2. Immunosensors Based on AuNPs

The working principle of an immunosensor relies on antigen-antibody binding and recent advances in nanofabrication-generated novel sensors with faster analysis time, enhanced sensitivity, and automated, low-cost, and low-volume per assay. More specifically, electrochemical immunosensors are well exploited for several research groups because they are easy to scale up, robust with enhanced sensitivity, and have improved limits of detection and quantification when analyzing small-volume samples [59,60].

In this way, combining traditional electrochemical sensors with AuNPs enables the development of high-sensitivity sensors; for example, Dequaire et al. used colloidal AuNPs to detect 3 × 10^−6^ μM by anodic stripping voltammetry and a lower detection limit was more sensitive than the gold standard detection by ELISA [61]. Liao et al. coupled square wave voltammetry and autocatalytic deposition to enhance the electrode surface by self-assembling AuNPs to detect up to 1.6 fM of IgG [62]. In addition to the well-established AuNP-decorated electrodes, the 3D functionalization of AuNPs sol-gel-matrix was developed by Chen et al. by self-assembling AuNPs with (3-mercaptopropyl)-trimethoxysilane with enhanced biocompatibility to detect human chorionic gonadotrophin with a detection range from 0.26 mIU up to 30 mIU [63]. This system improved the detection limit more than 100 times compared to the same sensing elements on flat electrochemical sensors. Including gold nanoparticles in the modified electrodes, the electron transfer between the transducer and biomolecules was enhanced, leading to improved bioanalytical devices. 

### 3.3. DNA Biosensors Based on AuNPs

DNA hybridization biosensors have increased in popularity among several groups recently. Developing DNA-based sensors needs high sensitivity during the transduction of the oligonucleotide, and electrochemical sensors combined with self-assembled AuNPs can provide new insights into the interface between DNA recognition and the signal-transduced element. Similar to the immunosensors, this can be conducted in a miniaturized platform with low volume/low cost and a reliable outcome. For example, Merkoçi et al. demonstrated the different strategies to couple AuNPs in DNA-based sensors [64,65]. The main strategies are the direct deposition of AuNPs onto the surface of a traditional genosensor (stripping voltammetry), the electrochemical detection of labeled-AuNPs, the AuNPs carriers of other AuNPs or electroactive labels, and the enhancement of AuNPs by using other metal ions such as silver. Using self-assembled AuNPs, Wang et al. developed a protocol to detect fM of DNA hybridization by capturing AuNPs in the target and performing anodic stripping voltammetry to quantify the metal tracer [66,67] electrochemically. The significance of this technology has been upgraded by establishing a more controlled synthesis protocol that is able to generate a range of sizes and shapes for AuNPs, enabling the appearance of novel and more sensitive electrochemical biosensors. Additionally, these methodologies could be applied to circulating DNA or proteins in the whole blood. 

### 3.4. Semiconducting Nanowires

Semiconduction nanowires possess unique properties that make them suitable candidates for developing novel electrochemical sensors. The main advantages of using nanowires include their mechanical strength, lightweight quality, ability to enhance a current, and their reduction in potential, resulting in sensors with an excellent limit range, sensitivity, and reliability [68]. 

Furthermore, the ability to generate high-quality materials with controlled specifications, such as diameter, length, and composition, has increased the focus on nanowires [68]. These nanostructures are typically synthesized using two main methods: the bottom-up approach, which involves self-assembling small-sized structures into larger ones, and the top-down approach, which involves reducing larger structures into smaller ones with multifunctional nanoscale features (Figure 2). One of the advantages of nanowires over carbon nanotubes is the refined control during synthesis, which allows for the external oxide layer to be used in a range of functionalization and blocking protocols [69,70].

Nanowires have been synthesized using various methods, such as crystallizing solid-state structures, template-assisted growth, anisotropic crystallographic structures, and liquid–solid interfaces to control their seed growth rate [71,72,73]. Among these methods, combining templates with self-assembly procedures is the most commonly used protocol in several research groups. Template-assisted methods provide a blueprint to direct the formation of the nanostructure. At the same time, self-assembly procedures enable the fabrication of nanowires with controlled specifications, such as diameter, length, and composition. By combining these methods, researchers have been able to synthesize nanowires with well-defined dimensions and tailored properties, making them attractive candidates for a wide range of applications in electronics, photonics, and sensing. For instance, nanowires synthesized using the template-assisted method have enhanced performance in electrochemical sensing applications due to their high aspect ratio, large surface area, and excellent electron transport properties [74,75].

The primary principle of the template strategy is to develop surfaces for nanowires using several top-down techniques such as electron beam evaporation, phase-shift optical lithography, sputtering, and molecular beam epitaxy [74,75]. Moreover, several types of templates generate nanowires, such as anodic alumina membranes, nanoporous anodic titania films, and anodic alumina membranes combined with the silicon substrate. Each one has its advantage; for example, the anodic alumina membrane provides the growth of well-ordered hexagonal cells, which allows for the enhanced yield of the diffusion process; the anodic alumina membrane combined with the silicon substrate can provide accurate control over the growth process, and the high purity of the synthesized product, and the nanoporous anodic titania film is well-known for its biocompatible properties [76,77].

The electrical properties of nanowires are an essential factor for their application. A range of features can influence conductivity, such as crystal quality, chemical composition, crystallographic orientation, wire diameter, and wire surface. In this way, several nanowire-modified electrodes have been reported in developing chemical sensors and biosensors [68,78].

### 3.5. Biosensors Based on Semiconducting Nanowires

One of the most common applications for chemical sensing using nanowires is the detection of hydrogen, ethanol, and hydrogen peroxide [68]. For example, ZnO nanowires combined with Pt were well exploited by Rout et al. through electrophoretic deposition in an anodic alumina membrane, resulting in the generation of a chemical sensor that was able to detect 1000-ppm ethanol [79]. Enzyme-based biosensors using nanowires are well-established at detecting hydrogen peroxide and glucose. For example, Au nanowires were electrodeposited in a polycarbonate membrane and used as the working electrode for glucose with a detection limit of 5 μM. A similar approach was made to detect hydrogen peroxide by coupling it with horseradish peroxidase to generate an amperometric sensor that was thirty times more sensitive than the conventional gold electrode [80].

Gold nanowires can also be used to detect cholesterol oxidase and esterase; the sensor developed by Aravamudhan et al. consisted of gold nanowires electroplated in a microfluidic platform by electroplating them in an alumina template that was able to detect cholesterol levels as low as 3 mM. In addition, a similar biosensor to detect cholesterol based on gold nanowires and Pt could detect cholesterol from 10 to 60 μM [81].

Ultimately, silicon and gold nanowires have been used to develop DNA sensors. For example, arrays of self-assembled silicon nanowires were synthesized by a complementary metal-oxide semiconductor that was compatible with technology, and a peptide nucleic acid capture probe based on silicon nanowires achieved a dependent resistance change upon hybridization to complementary target DNA with a detection limit of 10 fM, and another similar sensor was developed using arrayed gold nanowires for electrochemical DNA detection with a low limit of detection, suggesting that the 3D structures of the nanowires could enhance the diffusion at the microscale around these structures [68,82].

### 3.6. Self-Assembled Peptide Nanowires

Peptides are versatile for flexible frameworks due to their highly ordered self-assembly in 1D, 2D, and 3D structures. It can fold in motifs with nanosized structures such as monolayers, multilayers, fibers, micelles, and tubes, enhancing physio-chemical characteristics (Figure 3) [83,84]. However, there are also several disadvantages to using self-assembled peptide nanowires in these applications. Firstly, the mechanical strength of self-assembled peptide nanowires is relatively weak compared to other nanomaterials, such as carbon nanotubes or graphene. Secondly, the stability of self-assembled peptide nanowires is also a concern. They can be affected by changes in temperature, pH, and other environmental factors, which can cause them to break down or lose their structure. This can limit their lifespan and durability in specific applications. Thirdly, the production of self-assembled peptide nanowires is still relatively complex and expensive, which makes their mass production challenging. This hinders their commercial viability, as they are not yet cost-effective compared to other materials. Lastly, integrating self-assembled peptide nanowires into functional devices can be challenging due to their small size and unique properties. Developing reliable and efficient methods to incorporate them into functional devices remains a major challenge.

To overcome these limitations, several strategies have been proposed. For example, researchers have explored ways to reinforce the mechanical strength of self-assembled peptide nanowires, such as incorporating other materials or modifying their structure [85,86]. Additionally, efforts are being made to improve the stability of self-assembled peptide nanowires by developing new coatings or functionalizing their surfaces. For example, well-established protocols can quickly produce short peptides in a cost-effective and fast way. Excellent biocompatible materials can stabilize a range of molecules such as DNA, RNA, antibodies, enzymes, and other proteins [85,86]. While these have shown great potential as functional materials, some limitations need to be addressed. Nonetheless, ongoing efforts to overcome these challenges and the promising properties of self-assembled peptide nanowires suggest that they may play a significant role in future functional material applications.

### 3.7. Electrochemical Peptide Sensors

Electrochemical sensors based on peptides have been developed using a bottom-up synthesis by creating a self-assembled monolayer on the top of the transducer interface or by attaching the peptide as a recognition element on thiolate self-assembled monolayers [87,88]. Almost all the electrochemical methods have been used to measure the electron transfer of a peptide-based biosensor (e.g., cyclic voltammetry, square wave voltammetry, differential pulse voltammetry, alternating current, and electrochemical impedance spectroscopy). For example, several papers used peptide-based electrochemical biosensors to detect a range of molecules, such as plasmin, epidermal growth factors, prostate-specific antigen, colorectal carcinoma cells, and trypsin and HIV anti-p24 antibodies with a limit of detection as low as 1 pM with a linear range up to 1 nM [87,89,90,91]. 

Usually, for detection, the peptide is immobilized in a gold working electrode, and ferrocene or methylene blue Fc or MB are the primary redox probes. This peptide plays a role in the recognition element that engages in specific interactions with the target to combine with the three-electrode cell with reference and counter electrode; the principle of detection relies on the mediation of the process by the self-assemble peptide monolayer, which modulates the electron transfer between the redox tax and electrode surface. Therefore, the design of monolayers is essential for electrochemical sensing [92,93].

## 4. Optical Sensors

Among the most common types of biosensors are optical. They have the advantage of being very rugged, have high signal-to-noise ratios, a wide dynamic range, the real-time label-free detection of many substances, and are resistant to electromagnetic interference. Optical sensors are used in various applications, mainly for healthcare, biotechnology, and environmental applications. They are variants of photodetectors and detect electromagnetic radiation in the optical range. Optical sensors have many benefits over other sensor types, providing high speed, low price, and easy production [94,95].

### 4.1. Gold Nanoparticles

Gold nanoparticles (AuNPs) are particles that are less than 100 nanometers in size. These particles have unique optical and electronic properties that make them useful in various applications, including sensing, imaging, and therapeutics. The most important is their ability to absorb and scatter light efficiently. The optical properties of AuNPs depend on their size, shape, and surface structure. Their size determines the color of the gold nanoparticles [96,97].

The self-assembly of nanoparticles mediated by biomolecules has been widely used in the production of biosensors and in the formation of antibody–antigen complexes for the detection of various pathogens [98]. Among the most published studies using AuNPs is diagnosing viral diseases such as influenza. Influenza is a contagious viral infection that usually occurs in winter. It manifests itself with fever and respiratory symptoms and can progress in severe cases to a life-threatening level in high-risk individuals or groups [99]. Liu et al. developed a biosensor with colorimetric analysis for detecting influenza A virus (IAV) using antibodies functionalized with AuNPs. Initially, 13 nm AuNPs were modified with a monoclonal anti-hemagglutinin antibody (mAb) through a combination of ionic and hydrophobic interactions, followed by the addition of bovine serum albumin (BSA) to block the remaining spaces between the antibodies. This solution, containing AuNPs/mAb/BSA, showed a red color. After this step, an H3N2 (IAV) sample was added and incubated for 30 min. The AuNPs–mAb complex bound could be specifically arranged on the viral surface due to their specific antigen recognition, promoting aggregation and a shift in the absorption spectrum. The naked eye could detect a purple color. The limit of detection (LOD) was 7.8 hemagglutination units (HAU), presenting high specificity, accuracy, and good stability [100].

Using the conjunction of different nanoparticles, Basso et al. produced a hybrid nanomaterial formed from nanoparticles of γ-Fe_2_O_3_ (SAMN) and AuNPs to diagnose the dengue virus [101]. For the immunosensor production, SAMN was modified with 3-Mercaptopropiomic acid (MPA) for crosslinking with AuNPs, followed by conjugation with specific aptamers for the dengue virus. An aptamer was purchased commercially, and 5′-terminus was modified with the SH group for binding with AuNPs (Figure 4a). The concentration, sufficient to completely cover the surface of AuNPs, was 3.0 μM with an R^2^ value of >0.99. The immunosensor was characterized by transmission electron microscopy, FTIR spectroscopy, quartz crystal microbalance, and the LSPR technique. Positive samples were detected in real-time through colorimetric changes that were visualized with the naked eye and confirmed on UV-Vis graphics. The SAMN@MPA@AuNPs@aptamer complex detected only the four dengue serotypes (DENV 1–4) and not the Zika virus and yellow fever virus samples that were used as interferents, demonstrating high specificity and selectivity of the immunosensor. Compared to traditional antibodies, this new antigen detection methodology presented easy manipulation, low cost, simplicity of production, good bioconjugation yield, and large-scale production. The authors described it as an alternative diagnostic method.

Aptamers are oligonucleotides that can bind to a specific target molecule. They are usually composed of RNA or DNA and are often used in biomedical applications such as diagnostics and therapeutics. Some authors have reported that aptamers are more stable than antibodies and offer better functionality for protein detection. Faced with the new coronavirus pandemic that haunted the world in December 2019, numerous diagnostic methodologies emerged to detect the SARS-CoV-2 that causes COVID-19, including producing kits using AuNPs and aptamers. Aithal and collaborators described SARS-CoV-2 detection with aptamer-functionalized gold nanoparticles (nanoprobes). AuNPs in an aqueous solution were modified with a specific sequence of aptamers complementary to the spike protein in the viral membrane. The coagulant, MgCl_2_ salt solution (Salt M), was used to increase the signal and induce aggregation. These samples were then tested from cell cultures, and the detection tests were performed as follows: when the sample was negative without the presence of the spike protein. The addition of the Salt M neutralized surface charges on the AuNPs with aptamers, inducing their agglomeration and, when the sample was positive (with the presence of the spike protein) it bound to the aptamers and resisted agglomeration, enhancing steric stabilization and influencing its optical properties, which could be visualized through the Ultraviolet-Visible (UV-Vis) graph (Figure 4b). The nanoprobes detected 16 nM and higher concentrations of the spike protein in phosphate-buffered saline. With this methodology, 3540 genome copies/μL of inactivated SARS-CoV-2 were detected [102].

In the field of diagnosis in food safety, AuNPs have shown great promise as candidates for developing biosensors that can be used for common contaminant detection, such as mycotoxins, herbicides, pesticides, illegal additives, heavy metals, and pathogens. For example, malachite green (MG) is a cationic triphenylmethane dye that is used as a solid antifungal, antiparasitic, and antibacterial agent in fish farming industries. However, it can damage the immune function in humans, the nervous system, the digestive system, and the reproductive system, with carcinogenic, mutagenic, and teratogenic effects. Therefore, it can be classified as an illegal additive that is not authorized in many countries [103,104]. With this problem, Jia et al. developed a colorimetric aptasensor based on gold nanoparticles to detect MG in fish samples. AuNPs with 13 nm were modified with an aptamer sequence by the electrostatic interaction, which could protect AuNPs against aggregation. Then, the sample, either containing or not containing the MG and NaCl solution, was added. When the sample was positive for MG, aptamers bound to MG were detached from AuNPs, causing their aggregation to be induced by the presence of the salt, resulting in a final blue color. Alternatively, in the absence of MG, the aptamer did not detach from the surface of AuNPs, keeping them dispersed in a solution despite the presence of salt, resulting in a red color (Figure 4c). The experiment was effectively quantitative, with high sensitivity and specificity for MG detection. The linear detection range was found to be 20 to 300 nM, and the LOD was down to 15.95 nM [105].

Still, detecting allergens in food products is extremely important in food safety as undeclared traces of allergens in food can cause serious health problems in susceptible people [106]. The development of point of care (POC) for the detection of milk casein (CAS), bovine serum albumin (BSA), and egg chicken albumin (OVA), that commonly causes allergies in people worldwide, was described by Anfossi et al. The multiplex lateral flow immunoassay (xLFIA) was produced using AuNPs and silver nanoparticles (AgNPs) modified with antibodies against casein, ovalbumin, and hazelnut allergenic proteins to produce three differently colored specific probes [107]. Three proteins at levels as low as 0.1 mg/L in commercial biscuit samples were detected in a single strip. The colorimetric result was visualized in cyan/casein, yellow/ovalbumin, and magenta/hazelnut proteins (Figure 4d). The novel xLFIA provided a promising platform for lower cost and higher speed detection compared to traditional polymerase chain reaction (PCR) testing.
Figure 4Self-assembled nanomaterials using gold nanoparticles in the production of biosensors. (**a**) Hybrid nanomaterial formed from nanoparticles of γ-Fe_2_O_3_ (SAMN) modified with MPA solution, gold nanoparticles (AuNPs), and aptamers for the detection of dengue virus, Reprinted/adapted with permission from [101]. (**b**) Aptamer-functionalized AuNPs (nanoprobes) for SARS-CoV-2 detection; (**i**) Addition of the coagulant Salt M in an aqueous suspension with aptamer-functionalized AuNPs without the presence of the spike protein neutralizes surface charges on the nanoprobes, inducing their agglomeration; (**ii**) The suspension spike protein when present promotes steric stabilization without the formation of aggregates due to the spike protein bind with aptamers; (**iii**) UV-Vis graphs showing the wavelength shift in a colloidal suspension without the presence of spike protein, Reprinted/adapted with permission from [102]. (**c**) Colorimetric aptasensors for malachite green (MG) detection in a fish sample. AuNPs modified with aptamer sequence remained dispersed and showed a red color. With the addition of MG, binding with the aptamers occurred that detach from the AuNP surface, promoting the aggregation of AuNPs and changing the color to blue, Reprinted/adapted with permission from [105]. (**d**) Multiplex lateral flow immunoassay using nanoparticles for milk casein (CAS), bovine serum albumin (BSA), and egg chicken albumin (OVA) detection, Reprinted/adapted with permission from [107].
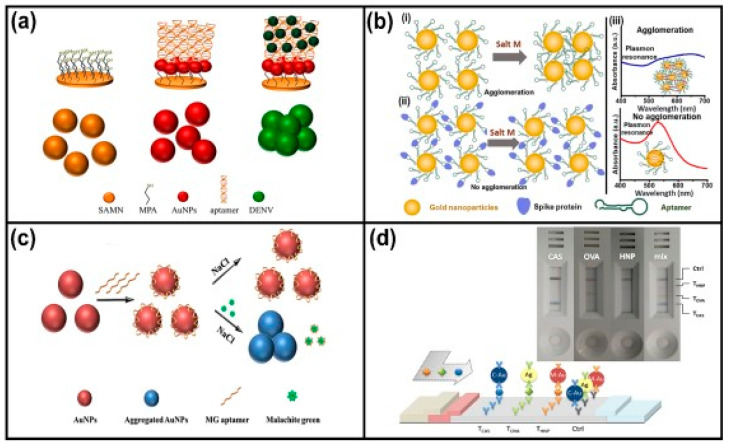


### 4.2. DNA Nanostructures

The DNA molecule transmits and stores genetic information in biological systems. Their essence of self-assembling, easy-to-design, and hybridization in the formation of nanostructures makes them a powerful tool for tracking and labeling biomolecules such as antibodies, nucleic acids, and proteins [108]. When self-assembled DNA nanostructures are loaded with intercalator dyes, the chromophores are protected, showing enhanced resistance to nuclease digestion. This principle increased energy transfer systems with tunable emission wavelengths [109]. Combining DNA nanostructures with fluorescence techniques could provide reconfigurable dynamics, allowing the visualization of a wide variety of cellular targets [110]. Using this approach, Tai et al. developed a snail-inspired DNA nanosensor for the real-time in situ detection of mRNA in living cells. This work was based on the structural composition of the snail, where the authors designed a DNA-based mRNA nanosensor (nano-SNEL) made up of a nanosensor and a nanoshell based on a fluorophore-quencher pair held close together by the complementary neck of a hairpin DNA. In the presence of the target mRNA, the fluorophore separated from the quencher to generate an intense fluorescence signal. The DNA molecule protected the sensor against enzymatic degradation and unspecific interactions, avoiding false positive signals. The final results showed that nano-SNEL detected the spatiotemporal distribution of mRNA transcripts in living cells without a cytotoxic effect [111].

DNA nanostructures are also widely described in the literature as tools to monitor the behavior of proteins, as their disordered expression level can be related to disease in most cases. Cell membrane surface receptors are transmembrane proteins that are embedded in the plasma membrane, transforming the extracellular signal to an intracellular signal through a signal transduction pathway. Such extracellular signals include growth factors, cytokines, hormones, neurotransmitters, cell recognition, and lipophilic signaling molecules. Therefore, it is necessary to monitor and regulate cell membrane surface receptors [110,111,112]. Based on this strategy, Liang et al. reported a nongenetic approach for imaging protein dimerization using C-Met (mesenchymal-epithelial transition receptor) by aptamer recognition and proximity-induced DNA assembly. In this work, the aptamer-specific recognition probe was used for the monomer receiver. The C-Met receptor as a model could bind to the hepatocyte growth factor (HGF) on the cell membrane. HGF activates the Met receiver by binding and promoting receiver dimerization. When receptor dimerization occurred, the dimeric receptors brought two aptamer probes into proximity, triggering DNA reassembly and causing the fluorescence recovery of probes. This technique allowed the visualization of two states (monomer/dimer) of a receptor protein on living cell surfaces in real time, allowing an easy investigation of the activation processes in signal transduction [113]. Wang and his collaborators developed a theranostic approach using a structure-switching aptamer. They triggered the hybridization chain reaction (HCR) on the cell surface for the real-time activation and amplification of fluorescence imaging and the targeting therapy of membrane protein tyrosine kinase-7 (PTK7). The LOD obtained was 1 pM, indicating a promising platform for early-stage diagnosis and the precise therapy of tumors [114]. Molecular self-assembly based on DNA hybridization has also been employed for the optical detection of nonenzymatically catalyzed artificial agents and the detection of HepG2 tumor cells in aptasensor [110,115,116].

### 4.3. Supramolecular Gels

The intermolecular arrangements of molecules or atoms are formed by different geometries and structures, which are directed by the functional groups and forces of interaction. In this context, supramolecular gels are available to construct nanostructures that can immobilize an enormous amount of solvent and can be used in drug delivery and tissue engineering to minimize inflammatory responses. Supramolecular gels are formed with homogeneous materials by treating a gelator solution with a stimulus normally from enzymes, light, or ultrasound [117,118]. The structure of these gels self-assembles through noncovalent interactions such as cation–anion electrostatic bonding, hydrogen bonding, van der Waals, metal-bonding coordination, ion-dipole, dipoles orientation, and π-interactions [117,118,119].

Fluorescent supramolecular gels have been increasingly used due to their excellent photophysical properties, particularly for their aggregation-induced emission behavior [120]. Pang et al. designed a naphthalimide (organic compound)-based fluorescent supramolecular gel containing an alkenyl group. This gel could be used to sense visually aliphatic and aromatic amines by measuring the change in the signal output with the enhancement of fluorescence intensity. The synthesis was performed by sonication in less than 2 min, producing an opaque and stable gel. However, there was a decrease in the fluorescence emission with the passage of the sonication time. The LOD was 1.30 × 10^−7^ M, and the correlation coefficient of R^2^ = 0.9976 [121]. Still, in the supramolecular gel approach, Feng et al. designed an ultrasound-triggered gelation approach to selectively solvatochromic sensors. This work synthesized two gelators with a terpyridyl group as an electron donor and a naphthalimide segment as an electron acceptor with amide bonds of different alkyl chains. When subjected to ultrasonication, these compounds solidified in organic solvents, visually indicating dimethyl sulphoxide (DMSO) from common organic solvents by the gel array in red. This supramolecular gel in DMSO could also detect H_2_O in samples, changing from red to yellow with an increasing water content from 0.5% to 10% [122].

### 4.4. Semiconductors

Changes in the optical properties of small molecules with the addition of different analytes can be used for detection in many areas. The analysis and quantitative detection of metal ions in water were performed using a molecular self-assembled colorimetric array based on pattern recognition algorithms. For example, mixing catechol dyes in water samples allowed for the colorimetric detection of metal ions as boronate esters were cleaved, resulting in a color change [123]. In this line of research, Sasaki et al. presented the sensing of a metal ion strategy using a molecular self-assembled colorimetric chemosensor array. Eleven different species of metal ions were detected simultaneously (Ni^2+^, Cu^2^, Zn^2+^, Cd^2+^, Co^2+^, Fe^2+^, Hg^2+^, Al^3+^, Pb^2+^, Ga^3+^, and Ca^2+^), and some metal ions were at sub-ppm levels. For this, catechol dyes such as pyrogallol red (PR), alizarin red S (ARS), bromopyrogallol red (BPR), and 3-itrophenylboronic acid (3-NPBA) were added to the water samples acting as a color manipulator accompanied by boronate esterification. When metal ions were bound to the dyes, they were simply detected as there was the induction of redshifts in the maximum absorption of the UV-vis spectra [124].

In this context, self-assembled chemosensor arrays are powerful tools to quantify multiple analytes in different types of samples. Optical probes were selected for chemosensor arrays using mathematical and chemical methods [125]. Smith et al. developed a fluorescence array for the detection of metal ions using a library-screening approach. This work used twelve fluorescent probes to identify nine metal ions in lake-water samples. The initial high throughput screening of the suite of metals identified the four best-performing probes with 100% accuracy. The other probes demonstrated different fluorescence behaviors upon adding the target metal ions. Although only four fluorescent probes efficiently detected the metal ions, their properties contributed satisfactorily to the optical response, allowing for the successful discrimination of different analytes [126]. Phenylboronic acid (PBA) derivatives and their boronate esterification were widely studied in another approach. They acted as Lewis acid metal centers in addition to forming cyclic boronate esters with cis-diol moieties of biomolecules in aqueous media, standing out in the production of sensory systems. Anzenbacher and coworkers described self-assembled colorimetric sensors using ZnII–DPA-attached PBA and catechol-type dyes to monitor ATP when hydrolyzed to pyrophosphate [127].

## 5. Conclusions and Future Perspective

Self-assembled monolayers (SAMs) have been extensively used in the fabrication of biosensors due to their ability to create a stable and controlled surface for biomolecular immobilization, which can help overcome limitations related to sensor lifetime and biological compatibility. However, there are several limitations that appear in terms of sensors and biosensors based on SAMs. One of the main limitations is the low sensitivity of these sensors, which is due to the limited surface area available for biomolecule immobilization. Another limitation is the instability of SAMs, which can result in the desorption or denaturation of the immobilized biomolecules. Additionally, the sensors’ lack of selectivity and specificity can result in false-positive or false-negative results. To address these limitations, several solutions have been proposed. One solution is to use SAMs in combination with other nanomaterials, such as nanoparticles, nanowires, or graphene, to increase the surface area available for biomolecule immobilization and enhance the sensitivity of the sensors [128]. Another solution is to modify SAMs with functional groups that can improve their stability and selectivity, such as thiol or amine groups [129]. Using molecular imprinting techniques can also improve the selectivity of the sensors by creating recognition sites that are specific to the target analyte. Developing new fabrication techniques, such as inkjet printing or microfluidics, can also enable the precise and scalable fabrication of SAM-based sensors and biosensors [130]. 

Additionally, a question that needs to be asked is whether using new functional nanomaterials is necessary to enhance the performance of a sensor or whether a sensor that does not incorporate a nanomaterial can perform equally well. As shown in this review, nanomaterials can improve the analytical figures of merit, such as sensitivity, selectivity, accuracy, and precision, as well as reduce assay time by overcoming diffusion limitations. However, the superiority of nanomaterials over macro or bulk materials can also stem from their dimension-related properties. For example, nanosized needles are necessary for measuring local analytes within a living cell: a feat that macro materials cannot achieve. Semiconducting nanowires have high electron transport efficiency and a high surface-area-to-volume ratio. Tunable electron transport properties along a confined longitudinal axis facilitate significant changes in their electrical properties upon small perturbations. However, the same material in larger dimensions cannot offer these properties. Therefore, the selection of nanomaterials for diagnosis and sensing must be considered case by case to determine their added value and functionality.

In summary, some limitations of sensors and biosensors based on SAMs can be overcome by combining SAMs with other nanomaterials, modifying SAMs with functional groups, using molecular imprinting techniques, integrating with advanced sensing technologies, and developing new fabrication techniques. These solutions have led to the development of sensitive and selective sensors and biosensors with wide-ranging applications in various fields, such as biomedical, environmental, and food analysis.

## Figures and Tables

**Figure 1 ijms-24-10819-f001:**
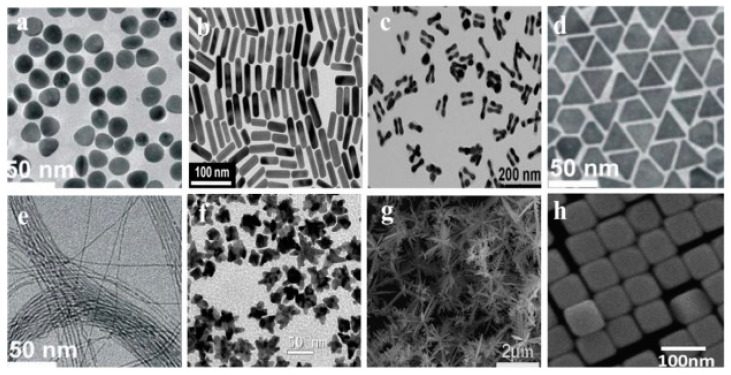
TEM images of several types of AuNPs structures: (**a**) Quasi-spheres (**b**) Nanorods, (**c**) Nanobubbles, (**d**) Triangular nanosprins, (**e**) Ultrathin nanowires, (**f**) Nanostars (**g**) Nanodentrides, and (**h**) Nanocubes, reproduced with permission from [46].

**Figure 2 ijms-24-10819-f002:**
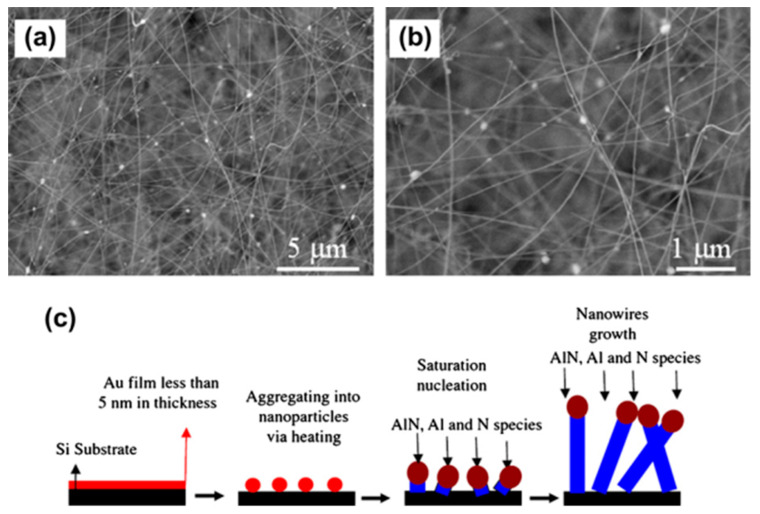
TEM images of nanowires (**a**) Low magnification (**b**) High magnification (**c**) Schematic of nanowire synthesis. Reproduced with permission from [46,68].

**Figure 3 ijms-24-10819-f003:**
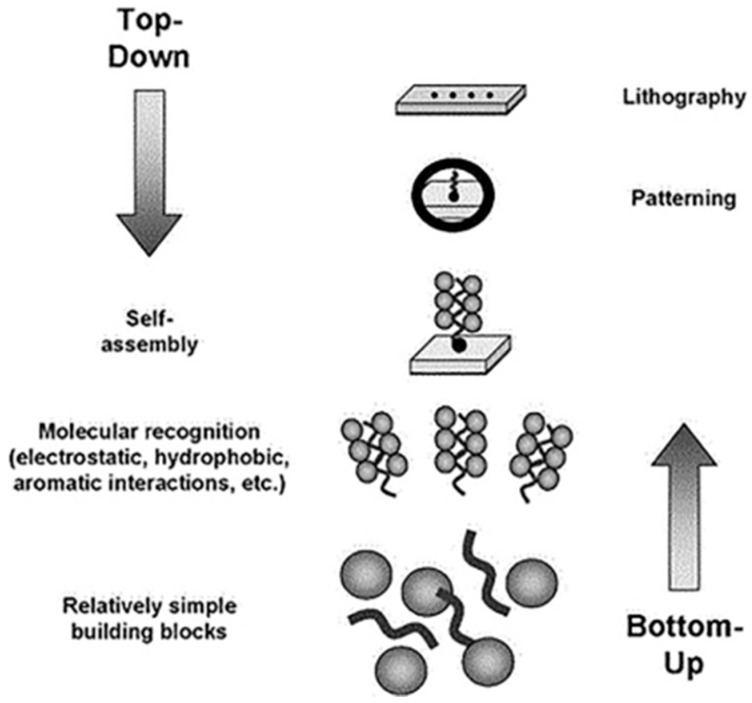
The scheme of peptide-based biosensor synthesis, reprinted/adapted with permission from [83].

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
