# Peer review of "Recent Advances in Functional Nanomaterials for Diagnostic and Sensing Using Self-Assembled Monolayers"

_ijms, 2023, doi:10.3390/ijms241310819_

Round 1

Reviewer 1 Report

Manuscript details:

Journal: International Journal of Molecular Sciences

Manuscript ID: ijms-2387587

Type of manuscript: Review

Title: Recent Advances in Functional nanomaterials for diagnostic and sensing using SAMs

Authors: Carolina R. Basso, Bruno P. Crulhas, Gustavo R. Castro, Valber A. Pedrosa*

   This review has been described on the functional nanomaterials for diagnostic and sensing using self-assembled (SAMs). These self-assembled nanomaterials have several potential applications, including medicine, electronics and energy, which providing a number of special properties. This review specifically mentions the sensors, such as immunosensor, DNA sensor, and optical sensor, using metal self-assembled nanoparticles or self-assembled nanowires. Furthermore, the review also showed supramolecular gels as examples of self-assembled structures through non-covalent interactions. Therefore, I think that it is an appropriate review for researchers in the fields of nanomaterials and biosensors. However, this review has contained a few indistinct points. Therefore, it deserves to be published after minor revision.

1. line 173: I think “e) star” is “f) star”. Additionally, I could not fine the explanation of e) in caption.

2. line 441: Where is Figure 4(f)? In addition, I could not fine the explanation of (d) in caption.

3. References: I think the reference in manuscript is old. Authors should refer to the most recent research in the authors’ manuscript. 

Minor

4. line 63: [14-16]. Period instead of comma.

Author Response

Manuscript IJMS (ISSN 1422-0067)
Thank you for considering our manuscript “Recent Advances in Functional Nanomaterials for Diagnostic and Sensing using Self-assembled monolayers.” for publication in RSC Advances. Overall we are thankful for the comments by Reviewers who carefully examined our manuscript, and we sincerely believe that the additional points in the revised version have been clarified now, thanks to the referee’s considerations. As a consequence, the revised version has considerably benefited from the referee’s comments.

Below we state each issue raised by Reviewers 1, and 2, and address/rebut each issue/comment raised, and indicate how and where we have changed the manuscript.

Referee: 1

1. line 173: I think “e) star” is “f) star”. Additionally, I could not fine the explanation of e) in caption.

Considering reviewer comments we rewrite the figure caption.

2. line 441: Where is Figure 4(f)? In addition, I could not fine the explanation of (d) in caption.

Considering reviewer comments we rewrite the figure caption. Instead of letter F is D.

3. References: I think the reference in manuscript is old. Authors should refer to the most recent research in the authors’ manuscript. 

Considering reviewer comments we have up date the reference .

Reviewer 2 Report

The article ~Recent Advances in Functional nanomaterials for diagnosis and sensing using SAMs~ is an interesting and very complex work. The subject is topical and can offer interesting perspectives to the scientific community. However, I believe that there is a major problem related to the organization of the article and some minor aspects that would bring improvements to the article. These aspects can be corrected/improved relatively easily, so that the article can be published in IJMS. See my suggestions below.

1. The introduction could contain several bibliographic references

2. The abbreviation ~SAMs~ must be explained right from the introduction.

3. Some sentences are quite long and could be divided into smaller, easier to follow sentences.

4. A more detailed explanation (1-2 sentences) of the self-assembly process and the forces behind it (only listed) would be helpful.

5. It would be useful to provide more information and own conclusions about the disadvantages and challenges associated with the use of nanomaterials in biomedical applications, such as toxicity and the difficulty of controlling the properties of nanomaterials during their synthesis and storage.

6. I believe that the entire section 2 (that is, almost the entire article) should be reorganized. Possibly, divided into several sections. There is a lot of information both about the nanomaterials used and about the biosensitive devices in the same section/subsection. This way of presentation is very difficult to follow and assimilate. Try to classify the information according to a certain criterion. For example: Section 2. Sensors, Section 3. Biosensors.... Or anyway you consider it more efficient so that the organization is much clearer.

7. Conclusions and future research perspectives should be improved. Highlight the solutions you propose to the limitations that appear in terms of sensors and biosensors based on SAMs.

8. Considering the large volume of information, I recommend enriching the bibliography with recent references.

Author Response

Manuscript IJMS (ISSN 1422-0067)
Thank you for considering our manuscript “Recent Advances in Functional Nanomaterials for Diagnostic and Sensing using Self-assembled monolayers.” for publication in RSC Advances. Overall we are thankful for the comments by Reviewers who carefully examined our manuscript, and we sincerely believe that the additional points in the revised version have been clarified now, thanks to the referee’s considerations. As a consequence, the revised version has considerably benefited from the referee’s comments.

Below we state each issue raised by Reviewers 1, and 2, and address/rebut each issue/comment raised, and indicate how and where we have changed the manuscript.

Referee: 2

1. The introduction could contain several bibliographic references.

Considering reviewer comments, we rewrite the introduction.

2. The abbreviation ~SAMs~ must be explained right from the introduction.

Considering reviewer comments, we include in the first line of the introduction.

3. Some sentences are quite long and could be divided into smaller, easier to follow sentences.

Considering reviewer comments, we have doubled checked the paper.

4. A more detailed explanation (1-2 sentences) of the self-assembly process and the forces behind it (only listed) would be helpful.

Considering reviewer comments, we rewrite this part.

5.It would be useful to provide more information and own conclusions about the disadvantages and challenges associated with the use of nanomaterials in biomedical applications, such as toxicity and the difficulty of controlling the properties of nanomaterials during their synthesis and storage.

Considering reviewer comments, we rewrite the conclusion.

6. I believe that the entire section 2 (that is, almost the entire article) should be reorganized. Possibly, divided into several sections. There is a lot of information both about the nanomaterials used and about the biosensitive devices in the same section/subsection. This way of presentation is very difficult to follow and assimilate. Try to classify the information according to a certain criterion. For example: Section 2. Sensors, Section 3. Biosensors.... Or anyway you consider it more efficient so that the organization is much clearer.

Considering reviewer comments, we reclassified the sections

7.Conclusions and future research perspectives should be improved. Highlight the solutions you propose to the limitations that appear in terms of sensors and biosensors based on SAMs.

Considering reviewer comments, we rewrite the conclusions

8. Considering the large volume of information, I recommend enriching the bibliography with recent references.

Considering reviewer comments, we rewrite the bibliography.

Round 2

Reviewer 2 Report

Congratulations to the authors!